# Payload State Prediction Based on Real-Time IoT Network Traffic Using Hierarchical Clustering with Iterative Optimization

**DOI:** 10.3390/s25010073

**Published:** 2024-12-26

**Authors:** Hao Zhang, Jing Wang, Xuanyuan Wang, Kai Lu, Hao Zhang, Tong Xu, Yan Zhou

**Affiliations:** 1State Grid Jibei Electric Power Company Limited, Tangshan 063000, China; wang_xuanyuan_1979@126.com (X.W.); zhang_hao_1980@126.com (H.Z.); 2Beijing Kedong Electric Power Control System Co., Ltd., Beijing 100192, China; wangjing19891110@126.com (J.W.); chn_lukai@163.com (K.L.); 3State Key Laboratory of Networking and Switching Technology, Beijing University of Posts and Telecommunications, Beijing 100876, China; zhy226080@163.com

**Keywords:** IoT network, network payload prediction, protocol state machine, cluster analysis

## Abstract

IoT (Internet of Things) networks are vulnerable to network viruses and botnets, while facing serious network security issues. The prediction of payload states in IoT networks can detect network attacks and achieve early warning and rapid response to prevent potential threats. Due to the instability and packet loss of communications between victim network nodes, the constructed protocol state machines of existing state prediction schemes are inaccurate. In this paper, we propose a network payload predictor called IoTGuard, which can predict the payload states in IoT networks based on real-time IoT network traffic. The steps of IoTGuard are briefly as follows: Firstly, the application-layer payloads between different nodes are extracted through a module of network payload separation. Secondly, the classification of payload state within network flows is obtained via a payload extraction module. Finally, the predictor of payload state in a network is trained on a payload set, and these payloads have state labels. Experimental results on the Mozi botnet dataset show that IoTGuard can predict the state of payloads in IoT networks more accurately while ensuring execution efficiency. IoTGuard achieves an accuracy of 86% in network payload prediction, which is 8% higher than the state-of-the-art method NetZob, and the training time is reduced by 52.8%.

## 1. Introduction

The Internet of Things (IoT) is developing rapidly and has huge potential in smart grids. The IoT network has achieved continuous progress and development due to its underlying network structures [1]. IoT networks have become important targets of various network attacks [2], such as viruses and botnets. In such an environment, network attacks can be found quickly by identifying the states of network payloads.

In IoT networks, individual nodes sometimes exhibit malicious behavior. Network attacks can be one of the reasons. For example, network viruses, botnets, and DDoS attacks can spread widely and quickly in an IoT network. Moreover, the number of nodes in an IoT network is huge, which leads to a wide range of security issues [3]. As long as one node is successfully attacked, the virus can be spread to neighbor nodes [4]. If a malicious node is detected early, it can be removed before spreading malicious payloads. The Mozi botnet is a typical malicious network, which could serve as an ideal research subject. As shown in Figure 3, the communication process between nodes cannot be accurately extracted. Current methods are unable to predict this type of network.

The existing detection methods usually identify malicious behavior by means of the state machines of network communication. According to the methods of constructing state machines, existing research can be divided into two types. One is based on network traffic logs [5,6]; the other is based on active interactions with target nodes [7]. To preserve more detail in the state machines, Antunes et al. [5] analyzed network traffic logs in detail, adding greedy algorithms and conservative alignment algorithms to traditional sequence comparison methods [8,9,10,11,12,13]. However, this approach led to the presence of redundant states in the state machine. Laroche et al. [6] constructed a time-sequential protocol dataset based on the response of the running protocol entity to discover an adaptable transition process of the protocol state. This method has less redundant state in the state machine, but will have an obvious impact on the real network environment. Zhang et al. [7] generated a large number of different protocol state sequences in advance, which made the accuracy of the constructed state machine higher, but this method needs to produce a large number of protocol data sequences, which takes a much longer time. 

The above method based on active interaction needs the deployment of long-term network traffic sniffers between active network nodes, which is challenging to implement in real-world scenarios. Meanwhile, this approach also requires prior knowledge of the network protocol specifications. In unknown networks, inability to obtain the network protocol specifications can result in inaccurate extraction of the protocol payload format. The method based on network flow logs requires extracting communication payloads from the logs and distinguishing between different types of payload states. For unknown network protocols, the number of payloads in different state types within historical network flows varies significantly due to network jitter. This leads to low accuracy in automatically classifying payload states, resulting in inaccurate construction of protocol state machines, which ultimately affects the prediction accuracy of malicious payload states between network nodes. In contrast to previous works, the malicious payload detection framework IoTguard proposed in this paper does not require the deployment of network traffic sniffers between network nodes. Furthermore, IoTguard does not rely on prior knowledge of network protocol specifications and is capable of predicting the states of payloads in unknown networks.

In this paper, a scheme for state prediction of payload networks based on network traffic is proposed. The state predictor of protocol payloads in this scheme can be used to classify the payloads of network traffic according to status type, and we name our predictor IoTguard. Because the protocol state machine of the TCP is relatively complex, we use a type of network traffic packet to represent the state machine of the network protocol. For example, the protocol state of the TCP network includes ACK, FIN, etc., while the protocol state of the botnet includes ping, find nodes, etc. Our method constructs a state machine of network traffic between different nodes, and clusters the payloads in network traffic to form clusters of multiple state types. Then, we compare the similarity between the state machine and clusters of payload state, and select the state clusters of payloads with the highest similarity to mark the payload state. IoTguard, which is trained on the dataset of payloads with state labels, can predict unknown network payloads more accurately in real time. We carry out our experiment on Mozi network payloads which were collected by us. It takes 10 days to collect 86k communication payloads between Mozi botnet nodes. 


**In summary, our contributions are as follows:**
(1)A scheme, IoTguard, for the prediction of payload state in unknown IoT networks based on network traffic is proposed. IoTguard is based on network traffic data and constructs a multi-classification model of payload state in an IoT network based on support vector machines(SVM). The constructed model can predict the payload state of an unknown IoT network.(2)An identification method for the state of network payloads without labels is proposed. The function of network payload identification is iteratively optimized based on state clustering of network traffic and state machine similarity. So, it can predict unknown network payload states and identify the unknown traffic payloads of an IoT network.(3)Experiments are conducted on Mozi real-time network traffic, which is a botnet which will often infect IoT devices. The results show that our method has higher accuracy than other methods, and can achieve earlier network warning. IoTguard achieves an accuracy of 86% in network payload prediction, which is 8% higher than the state-of-the-art method NetZob [11]. Compared with NetZob, IoTguard reduces the training time by 52.8%.


The structure of this paper is as follows. Section 2 provides background information, including a review of related work in the field of network payload prediction and state machine construction. In Section 3, we introduce the proposed approach of IoTguard. Section 4 presents the experimental setup, followed by the results and discussion in Section 5. Finally, Section 6 concludes the paper and discusses potential future work.

## 2. Background

### 2.1. Related Technologies

#### 2.1.1. State Machines

A state machine [14] is a mathematical model that represents a finite number of states and behaviors such as transitions and actions between these states. Taking the TCP state machine as an example, during TCP’s three-way handshake and four-way wave, each transmission will go through the process of connection, transmission, and closure. During the communication process between the two parties, the status of the TCP protocol is different, for example, LISTEN, SYN-SENT, etc. These states will be converted during communication. When a server intends to establish a TCP connection, its state transitions are depicted as shown in Figure 1.

#### 2.1.2. State Machine Similarity

State machine similarity can measure the similarity of two state machines. Based on established fact, similar state machines have similar eigenvalue distributions [15]. Eigenvalues are extracted from transition matrices of a state machine, which capture the structural information of the state machine. When the eigenvalue distributions of multiple state machines are very close, their structures are considered to be similar, so the similarity is high. 

Suppose we have a transition matrix of a state machine, and its eigenvalue set is Characters={λ1, λ2, …,λ}. These eigenvalues can describe the structural attributes of the state machine. The average of these eigenvalues is expressed mathematically as follows in Equation (1):(1)$$Aver=\frac{1}{n}\sum_{i=1}^{n}\lambda_{i}$$

The standard deviation of the eigenvalues σ is calculated as the difference between each eigenvalue and the mean. The formula for standard deviation is as follows in Equation (2):(2)$$\sigma =\sqrt{\frac{1}{n}\sum_{i=1}^{n}{\left(\lambda_{i}-Aver\right)}^{2}}$$

When the standard deviation of the eigenvalues is small, the distribution of eigenvalues is more concentrated, and state machine similarity is higher. Therefore, the reciprocal of the standard deviation is taken as the similarity, and state machine similarity is defined as in Equation (3):(3)$$similarity=\frac{1}{\sigma }$$

#### 2.1.3. Hierarchical Clustering

Hierarchical clustering [16] analyzes data at different levels based on the similarity between clusters, thereby forming a tree-shaped clustering structure. The hierarchical clustering algorithm assumes that each sample point is a separate cluster and then finds clusters with higher similarity in each iteration to merge. This process is repeated until the preset cluster is reached. Hierarchical clustering is an unsupervised algorithm that progressively builds clusters based on data similarity. It does not require predefining the number of clusters and allows for dynamic parameter adjustments.

#### 2.1.4. Payload State Cluster 

Payload state clustering involves classifying payloads with different states using clustering algorithms. For example, the hierarchical clustering algorithms mentioned earlier are used to cluster payloads. Figure 2 following illustrates the two clusters formed after clustering two types of payloads, corresponding to the ping and find nodes states, respectively.

#### 2.1.5. SVM

A SVM [17] is a classic supervised learning algorithm used to solve binary and multi-classification problems. The core principle is to classify by finding an optimal hyperplane in the feature space with the largest interval.

### 2.2. Related Work

#### 2.2.1. Payload State Prediction

Quoc-Dung Ngo et al. [18] employed static features of payloads to identify state in real-time network traffic, achieving high timeliness. However, existing approaches require prior knowledge of network protocols, making them inapplicable to protocols without predefined specifications. 

The scheme for predicting network payload state based on network traffic extracts the data payload by analyzing the characteristics of the transmitted payloads. It then constructs a state machine of network protocol according to the transmission sequence of packets, and we use payload categories as the state in the protocol state machine. Andrea Fioraldi et al. [8] replayed payloads from network traffic to remote nodes to obtain payloads for the next state phase. They differentiated payload state based on the differences between payloads in two phases. This approach interacted with actual network nodes and could affect them, leading to insufficient stealth. Meng et al. [9,10] used a multi-sequence alignment algorithm based on Hidden Markov Models to infer payload formats in network traffic and automatically identified state fields within the payloads. They constructed protocol state machines based on these state fields within a single TCP connection. Bossert et al. [11] adopted an inference method for protocol format based on sequence format alignment. Firstly, the payload format is inferred, then, the payload is classified into various state types, and finally, the state machine is inferred based on message sequences. Since many network protocol payloads have state fields that are relatively concealed, this method required a large amount of historical data to infer the state types of protocol payloads.

#### 2.2.2. Protocol State Machine Construction

There are two main ways to construct a protocol state machine. One is to passively analyze the state machine based on network traffic [12], and the other is an active interaction by sending requests to the target program and receiving responses [13]. 

Antunes et al. [5] applied a greedy algorithm and a conservative alignment algorithm to the traditional sequence alignment method, enabling the method to retain non-frequent information. New nodes are created for information not present in previous sequences, preserving as much of the state machine’s node information as possible. This is a passive method of state machine analysis, with the advantage of retaining more detailed information in the state machine. However, its drawback is that it generates numerous states in the state machine, potentially producing multiple sub-states for the same actual state.

Laroche et al. [6] proposed real-time interaction with the program running the protocol entity, constructing a dataset of time-sequential protocols based on the obtained request–response correspondence, and then comparing multiple protocol datasets to find a more adaptable transition process of protocol state. Zhang et al. [7] generated a large number of different detection sequences of protocol states in advance, and interacted with the target program according to the generated sequence each time to filter out the correct protocol state machine. This method mainly proposes an improved QSM method [19], which significantly improves the accuracy of constructing the protocol state machine. The accuracy of constructing a state machine using this method is high, but the process of inferring the state machine protocol data sequence takes a long time. When the number of known data payloads is large, a large number of protocol data sequences will be constructed, and the time cost of trying these sequences sequentially is high.

## 3. Materials and Methods

### 3.1. Problem Formulation

Existing methods of payload state prediction typically rely on the deployment of network sniffers and require prior knowledge of network protocols. However, these methods are ineffective in predicting unknown payload states. This paper proposes a novel approach that enables the prediction of unknown payload states without requiring prior knowledge or the deployment of network sniffers.

### 3.2. Scheme Design Motivation

Compared with the network traffic of industrial standardized networks (hereinafter referred to as conventional networks), there are more nodes which communicate with each other in malicious IoT networks than in conventional networks, and the timing stability of datagrams transmitted between malicious network nodes is worse than between conventional networks. After sending a datagram in a regular network, the response datagram is rarely lost or duplicated, but these two situations are very common in malicious networks.

Taking the Mozi botnet [20] as an example, the Mozi botnet often infects IoT devices, such as network gateways and digital video records. Mozi attacks the IoT device through weak telnet passwords and unpatched IoT vulnerabilities. After a single botnet node starts running, it communicates with 238 other different Mozi nodes within 30 min and transmits the information of mutual recognition and synchronization between nodes. The Mozi botnet is a typical IoT botnet, where the communication between nodes relies on the UDP protocol. When communicating between nodes, problems such as out-of-order datagrams and failure to receive responses are often encountered. As shown in Figure 3, these problems result in the inaccurate extraction of the communication process between nodes. However, most of the application layer protocols of normal industrial standardized networks use the TCP protocol in the transport layer and adopt a mechanism of datagram label confirmation.

In Figure 3, the red row is the request datagram sent by the local node to the peer node, and the blue row is the response datagram sent back by the peer node to the local node. The first column is the original data of the network byte stream, the second column is the text information corresponding to the byte stream, and the third column is the payload state type obtained after manual reverse engineering of Mozi binary samples and traffic analysis. The ping and find nodes lines in red font in Figure 3 reflect the influence of the network environment fluctuations during the communication process between Mozi botnet nodes, resulting in the problem of out-of-order and repeated UDP datagrams in requests and responses.

Due to the stealthiness and unknown characteristics of malicious network attacks, the network communication protocols between nodes are different from conventional network protocols. Most of them are custom-developed communication protocols, or they are tailored and transformed from existing network protocols. Therefore, it is impossible to directly analyze the protocol type in application layer from the network traffic level. For example, the commonly used network analysis software Wireshark-3.4.6.0 can only identify the Mozi botnet communication traffic analysis as the transport layer UDP protocol, but cannot further analyze its application layer protocol, as shown in Figure 4.

According to the protocol data transmission shown in Figure 3, the corresponding communication process between a pair of nodes can be delineated. The ping and find nodes marked by red lines in Figure 5 correspond to the two rows highlighted in red font in Figure 3. Specifically, after local node A sends a ping payload, peer node B sends a ping response datagram back to the local node, and then node A sends a repeated ping payload to the peer node. This is because node A times out while waiting for the PING response payload, so node A resends the PING payload. At this time, it happens to receive the ping response payload returned by node B. Next, node A sends two consecutive find nodes to node B to find neighbor nodes, but only receives one response from node B to the find nodes command, and the other find nodes command is lost due to network reasons. This problem of out-of-order UDP datagrams and failure to respond due to the network environment leads to inaccuracies in existing approaches which apply Hidden Markov Models [9,10].

IoTguard uses state transition diagrams to represent state machines. Although these transition diagrams are not strictly modeled in the formal framework of state machines, they effectively and concisely illustrate the logical relationships between states. The protocol state machine of the above process is shown in Figure 6. In the first step of the state machine, the ping state is transferred to the ping response. In the second step, the ping response state is transferred back to ping. In the third step, the ping state is mistakenly transferred to find nodes. In the fourth step, the find nodes state jumps back to itself. In step five, the find nodes state transferred to the find nodes response state.

The state machine of the Mozi communication protocol is incorrect. If this state machine is used as the basis for adjusting the clustering parameters of payload state, the number of clusters of payload state formed after clustering will be inaccurate, and there will appear a situation where multiple payload state clusters correspond to the same actual state type, or a large number of payloads of different state types will appear in a single payload state cluster. Part of the clustering is shown in Figure 7.

### 3.3. Prediction Scheme

This section designs a prediction scheme for payload states in an IoT network based on the clustering algorithm and the comparison of state machine similarity. In a malicious network node, any node will communicate with most other malicious network nodes in a similar mode. This is a feature that a conventional network node does not have. Therefore, this feature can be used for similarity comparison to dynamically adjust the hyperparameters of clustering. So, it can improve the clustering accuracy of unknown network payload state, thereby improving the prediction accuracy of the network payload state. This solution does not require pre-analysis of binary samples, nor does it require prior knowledge of the network to obtain accurate communication protocol logic. It only predicts the network payload states based on network traffic.

#### 3.3.1. Overview

The prediction scheme for payload states in an IoT network includes three major steps: network payload separation, payload state extraction, and generation of the payload state predictor. 

In the first part of IoTguard, network payload separation is responsible for extracting the traffic between the local network node and each other remote node and network payloads in the application layer from the unknown network node’s traffic. The input is the pcap file of network traffic *all_node_traffic* between network nodes. Through the extraction algorithm of network payloads in the application layer *PayloadExtractor*, the pcap file set of communication traffic *node_traffic_n* local node and each remote node and the set of all payloads during communication *payload_from_traffic* are obtained.

In the second part of IoTguard, payload state extraction is responsible for producing accurate payload state classifications in captured network traffic. The inputs are *node_traffic_n* and *payload_from_traffic*. By constructing a protocol state machine for the communication process between each pair of nodes and dynamically optimizing the state classification accuracy of the payloads through the comparison algorithm of state machine similarity, the output is a list of payloads divided by state *payload_state*. 

In the third part of IoTguard, the generation of the payload state predictor is responsible for generating a payload state predictor in the network. The input is *payload_state*. After training a multi-classification model by state category labels, a multi-classifier is applied to predict the state category of network payloads. The output is the payload state predictor, referred to as *payload_state_predict*.

As shown in Figure 8, the network traffic is history data obtained by performing network traffic sniffing and packet capture on a normally operating network node. The network traffic stores the communications between the local network node and other nodes on the same network. The communication datagrams include source and destination node addresses, protocol type in the transport layer, original payload data in the application layer, datagram timing, and other information.

The application layer payload list in Figure 8 only includes application layer payloads for communication between the local network node and other nodes, and does not include meta-information such as timing and node addresses. The traffic between these nodes is stored in a set of pcap files, each of which only contains the communication between the local node and a single remote node. The payload list divided by state comprises multiple different payload list files, and each file is named with a state type number as a label.

After undergoing the extraction process of payload states, the list of application layer payloads and the collection of traffic between the local node and other nodes are transformed into a set of application layer payloads corresponding to each state. These state-divided payload lists serve as inputs to the generation process of the payload state predictor, which is used to train and generate the payload state predictor.

#### 3.3.2. Network Payload Extraction

This step uses the extraction algorithm of network payloads in the application layer to extract two parts of the content from the network traffic between network nodes. The first part is the record of communication transmissions between the local network node and other different nodes on the same network, and the record is sorted by the time order of transmitting datagrams. The second part is a list of network payloads in the application layer generated by the local network node when communicating with other nodes. The output of network payload extraction is shown in Figure 9.

As can be seen in Figure 9, the input of network payload extraction in the application layer is the pcap file of the communication history traffic *all_node_traffic* between the IoT network nodes. The output is in two parts:

The first part is the historical traffic *node_traffic_n* between the local network and different remote network nodes; each set of historical traffic data is stored in the form of a pcap file, and each set of historical traffic data contains all communication data between the two network nodes, including transmission timestamp, transmission sequence information, IP address, port number, etc. These data will be used for reverse engineering of the protocol state machine.

The second part comprises the network payloads of the application layer in the historical communication traffic between all the network nodes and forms a list. The table only contains the original binary data of the application layer payloads *payload_from_traffic*, and it does not include the addresses of the sender or the receiver, datagram timing, or other information. These data will be used for clustering to obtain network clusters of payload state. Network adapter extraction in the application layer is shown in Algorithm 1. Assuming the number of packets in *all_node_traffic* is m, the time complexity of Algorithm 1 is O(n).

The extraction algorithm for network payloads in the application layer PayloadExtractor takes the network traffic data *all_node_traffic* between network nodes as input. The output contains two parts. One is the pcap file set of communication traffic *node_traffic_n* between the local node and each remote node, and the other is only the original binary data of the network communication in the application layer *payload_from_traffic*.

Extracting the network traffic between each pair of nodes (lines 1 to 16): Firstly, we traverse all datagrams in the network traffic *all_node_traffic*. According to the IP address of the remote network node in each datagram, the datagrams are classified into independent records in chronological order. Secondly, these independent records are generated as individual pcap files which are used as the communication file collection of the traffic pcap *node_traffic_n* between the local node and each remote node.

Extracting the network payloads in the application layer (lines 17 to 22): Firstly, we traverse all datagrams in the network traffic *all_node_traffic*. Secondly, we extract the binary payload data in the application layer in these datagrams and add them to the payload list.
**Algorithm 1:** Extraction algorithm of network payloads in application layer PayloadExtractor
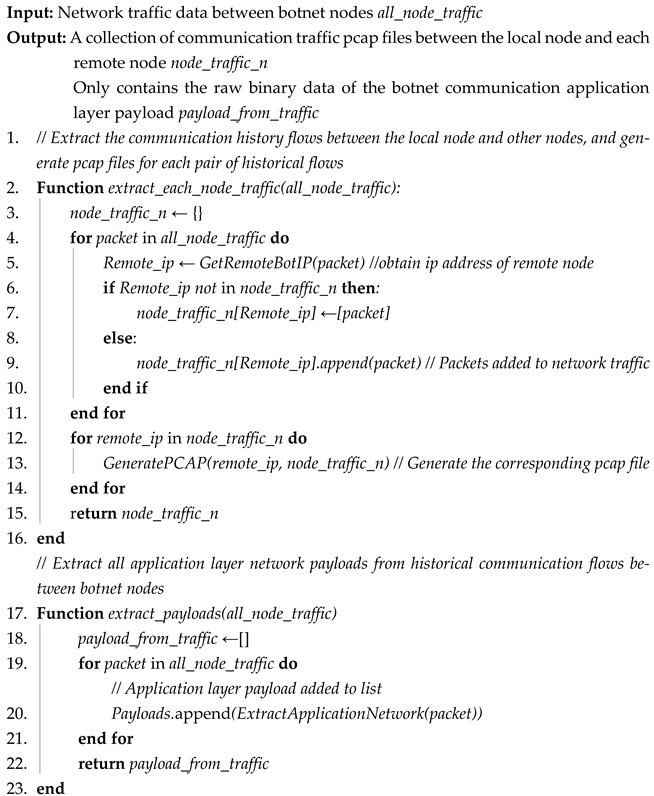


#### 3.3.3. Payload Status Extraction

This step uses the clustering algorithm [21], state machine construction, and similarity comparison method to extract the classification clusters of the original payload state from the communication payload between nodes. This step consists of three parts. As shown in Figure 10, the first part is to use the clustering algorithm to cluster all payload sets *payload_from_traffic* during communication. The second part is using the clustering results and *node_traffic_n* to construct a state machine for the communication process between a pair of nodes. The third part is comparing the similarities of multiple state machines, obtaining similar results, and producing clustering results when the similarity of multiple state machines is the highest.

As shown in Figure 10, the input for the extraction process of payload state is the set of all payloads *payload_from_traffic* during communication and the set of pcap files *node_traffic_n* of the communication traffic. The output is the state clustering results which use payload lists divided by state as the original payloads. The payload clustering process firstly extracts features of all input payloads, and then uses the hierarchical clustering algorithm [22] to cluster the extracted features. The clustered clusters of payload state will be used in the second step. We construct a protocol state machine of the network. This process is shown in Figure 11. The third step is to compare the state machine similarity of all constructed state machines to obtain the state machine similarity. The fourth step is adjusting the parameters of hierarchical clustering and performing re-clustering.

As shown in Figure 11, an autoencoder [23] is used to extract features from the input raw payloads. The dimensionality reduction data from these extracted features are used as the input for a hierarchical clustering algorithm. The hierarchical clustering algorithm performs unsupervised learning on the input feature vectors, classifying them into multiple clusters. Based on the original payloads in the application layer corresponding to the feature vectors stored in each cluster, a payload list divided by states, called *payload_states*, is generated.

The input in constructing the state machine is the payload list *payload_states* divided by state and the set of pcap files *node_traffic_n*. This process uses the datagrams and their timing between each pair of nodes in *node_traffic_n* to construct the state machine and uses weights to mark the transfer direction between states in the state machine, and the output is the state machine corresponding to the communication traffic between the local node and each remote node. The state in each state machine is named after the cluster number of *payload_state*. The process of constructing a state machine is shown in Algorithm 2. Assuming each *node_traffic* contains n packets and *node_traffic_n* includes traffic for k nodes, the time complexity is O($k\cdot n^{2}$).
**Algorithm 2:** Constructing a state machine algorithm
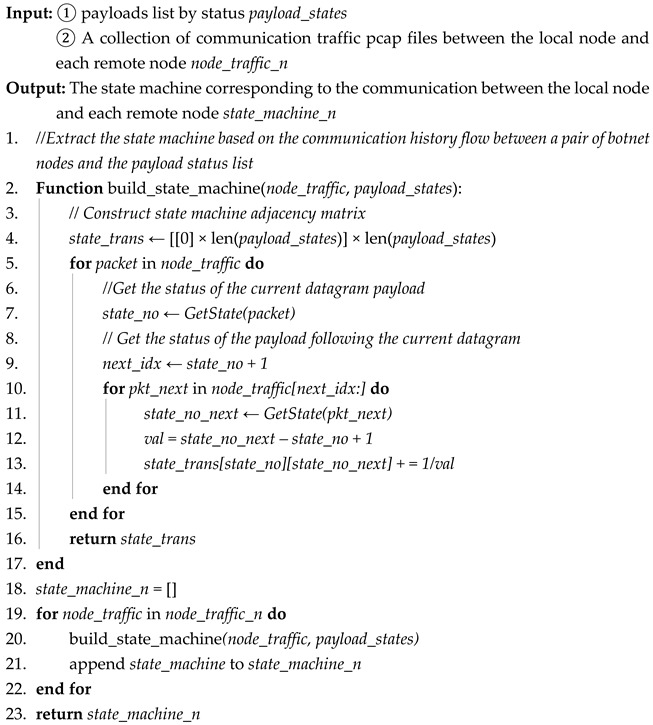


The steps of Algorithm 2 are as follows:

Initializing state machine parameters (line 3): Firstly, we construct the protocol state machine adjacency matrix, which represents the possibility of state transition in the state machine in the form of weights.

Obtaining the payload state type in the application layer of the starting node in the state machine (lines 6 to 8): We traverse each message in the *pcap* file of the communication traffic between nodes, and we obtain the clustering results for the application layer payloads in each message. We use the cluster number obtained by clustering as the state type of the payloads.

Constructing a protocol state machine with the starting datagram as the header (lines 5 to 15): For the payloads in the datagram after the starting datagram, we use its cluster number in the clustering result as the state type; then, the reciprocal of the interval between the subsequent datagram and the initial datagram is used as the state transition weight. The relationship between the sequence number $Idx_{m}$ of state *m*, the sequence number $Idx_{n}$ of state n, and the conversion weight $Weight_{trans\_mn}$ is as shown in Equation (4), where n is greater than *m*.
(4)$$Weight_{trans_{mn}}=\frac{1}{\left(Idx_{n}-Idx_{m}+1\right)}$$

Constructing a state machine for each inter-node pair of network traffic instances (lines 18–23): We repeatedly use the method *build_state_machine* to construct a state machine for each inter-node pair of network traffic to generate all state machines *state_machine_n*.

The third step is comparing the similarity of the state machines *state_machine_n* generated in the second step. Firstly, we calculate the eigenvalues of the state transition matrices corresponding to all state machines. Secondly, we calculate the standard deviation of these eigenvalues, and use the standard deviation as the similarity of all state machines: *similarity_machine*. This process is shown in Algorithm 3. Assuming the number of state machines is n, and the dimension of the state transition matrix for each state machine is k, the time complexity for calculating the state machine similarity is O($n\cdot k^{3}$).
**Algorithm 3:** State machine similarity comparison algorithm
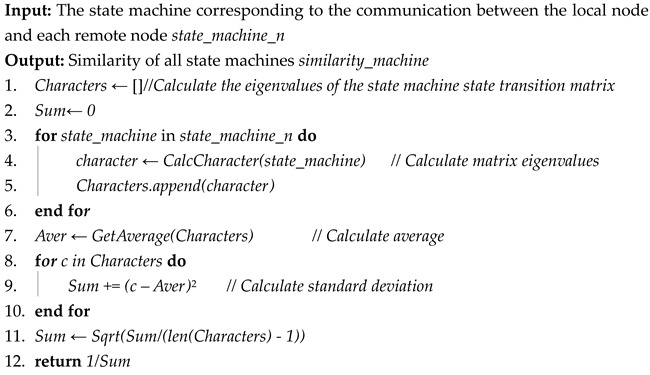


The input of the comparison algorithm of state machine similarity is the state machine *state_machine_n* corresponding to the communication between the local node and each remote node. Its expression is n adjacency matrices, and the output is the *similarity_machine* obtained by similarity comparison of all protocol state machines.

The process of Algorithm 3 is as follows:

Initialization (lines 1 to 2): We initialize the state machine state transition matrix eigenvalue list Characters and initialize the denominator of the state machine similarity to 0.

Calculating the eigenvalues of all state machines (lines 3 to 6): We traverse the state transition matrices of all state machines, calculate the eigenvalues of each matrix, and add these eigenvalues to Characters.

Calculating similarity (lines 7 to 12): First, we calculate the average value of eigenvalue Aver in Characters. Then, Aver and each eigenvalue are used to calculate the standard deviation of the overall eigenvalue according to the standard deviation formula. The reciprocal of this standard deviation is then used as the state machine similarity.

Because the communication patterns between a single network node and other nodes are similar [24], multiple state machines are the same in an ideal state, that is, the similarity of the state machines is 1. The higher the similarity between state machines is, the more accurate the classification of state categories used to construct the state machine.

As shown in Figure 10, after calculating the similarity of the state machine once, the extraction process of payload state will linearly adjust the parameters of the hierarchical clustering in the second part to increase the threshold parameter of the hierarchical clustering by 0.0005. We repeat the process in Figure 10 five times, and the hierarchical clustering results are obtained as a payload list divided by states *payload_states* when the state machine similarity is the highest. This payload list is input into the fourth step to train the payload state predictor.

#### 3.3.4. Generating the Payload State Predictor

This step uses a classifier of the SVM to construct a state number classifier for the payload, and obtains the correspondence between the state number and the payload in the application layer from the payload list *payload_states* divided by state; it uses the SVM to train Cntst binary classifiers, and the value of Cntst is the total number of states. Each binary classifier is only used to classify whether the input payloads in the application layer belong to a certain state number. During training, the payloads in the application layer of a single state are used as a positive example, and all other payloads in the application layer are used as negative examples. During prediction, all trained binary classifiers are used to predict samples at the same time, and the state number of the binary classifier with the highest confidence in the prediction results is taken as the state prediction result of the corresponding payloads. This process is shown in Figure 12.

Each binary classifier in Figure 12 only uses the payloads in the application layer which just has one state as a positive example, and the remaining samples are all negative examples for training, so that a single binary classifier can only predict the payload with one state. There are Cntst kinds of payloads in the input payload list divided by state, corresponding to Cntst binary classifiers. The process for predicting the state of a payload in the application layer is illustrated in the lower part of Figure 12. After the payloads are input into Cntst, all binary classifiers generate their own confidences of prediction results, and these confidences are summarized into a one-dimensional array. The state numbers of the payloads are obtained by selecting the maximum value in the one-dimensional array of confidence levels.

## 4. Results Evaluation and Discussion

### 4.1. Experimental Setup

#### 4.1.1. Problem Statement

To validate the effectiveness and practicality of the proposed scheme for predicting the payload state of unknown IoT networks, this section conducts related comparative experiments to answer the following questions:

RQ1: Is it effective to dynamically adjust clustering parameters by comparing the similarity of protocol state machines? (Answer in Section 4.2, corresponding to A1)

RQ2: What are the optimal parameters of each parameter in the prediction scheme for payload state in an IoT network based on historical traffic? (Answer in Section 4.3, corresponding to A2)

RQ3: Is the training model of the payload state predictor reasonable? (Answer in Section 4.4, corresponding to A3)

RQ4: Is the prediction scheme IoTguard based on historical traffic better than the existing prediction scheme? (Answer in Section 4.5, corresponding to A4)

#### 4.1.2. Experimental Environment

The implementation of this study was carried out on the Ubuntu 20.04 system equipped with Python 3.9, Wireshark 3.4.6.0, and VirtualBox 7.0; the hardware computing resources were an Intel i7 10750H, 16 G memory PC and a public network server static IP.

#### 4.1.3. Dataset

This study used the existing sample of Mozi botnet binary and ran the Mozi sample on a server visible on the public network. The running time of a single binary sample was about 10 days, and a total of 86k communication payloads between Mozi botnet nodes were collected.

### 4.2. Experimental Evaluation of Parameters Adjusting

Evaluation index: We used the ARI(Adjusted Rand index) [25] to evaluate the clustering effect.


**A1: Effectiveness Evaluation Experiment (Answering RQ1)**


The purpose of this experiment was to evaluate the effectiveness of the hierarchical clustering method in predicting unknown network payload states. This was done by comparing the clustering outcomes in two cases: one with dynamically adjusted clustering parameters, and one without. To account for the inherent instability of the clustering algorithm, each scenario was tested 20 times to draw generalizable conclusions.

Experimental results and analysis: Figure 13 shows a comparison of the effects of clustering between adjusting hierarchical clustering parameters and not adjusting hierarchical clustering parameters.

It can be seen from the experimental results that the clustering effect without adjusting the hierarchical clustering parameters exhibits significant fluctuations in 20 experiments; the clustering performance with parameter adjustment shows more consistent results with no major variations. The ARI [25] coefficient of the clustering results obtained by hierarchical clustering after dynamically adjusting parameters is higher than that by hierarchical clustering without adjusting parameters. Specifically, the dynamically adjusted method outperforms by 0.056 in the worst scenario and by 0.123 on average.

### 4.3. Comparative Experimental Evaluation of Parameter Selection

Evaluation metrics: We evaluated autoencoder training speed using model training time, and used cosine similarity cos_sim [26] to evaluate the effect of feature vector extraction with the autoencoder. The cosine similarity is shown in Equation (5); we define vecA as the original vector and vecB as the output vector of the decoding part of the autoencoder. We used the adjusted Rand coefficient ARI to evaluate the clustering effect.
(5)$$cos_{sim}=\left(vecA\cdot vecB\right)/\left(\left|vecA\right|\cdot \left|vecB\right|\right)$$


**A2: Comparative Evaluation Experiment (Answering RQ2)**


In order to evaluate the training speed of the autoencoder, the training times were compared for autoencoders with 3, 4, and 5 layers in both the encoder and decoder. Additionally, the cosine similarity between the input payload vectors and the decoder output vectors was calculated for autoencoders with different layer configurations to assess data loss and reflect the feature extraction effectiveness of the autoencoder.

Due to the inherent uncertainty in each training use of the autoencoder, 20 training iterations were conducted for autoencoders with different numbers of layers to obtain more generalizable experimental results. The number of layers in the encoder and the hyperparameters for each layer are shown in Table 1, with the decoder having a symmetric structure to the encoder.

For hierarchical clustering, the ARI of the clustering results was compared across 15 different initial threshold parameters to evaluate the impact of these parameters on clustering performance and identify the optimal initial parameter.

Experimental results and analysis: Figure 14 shows the training time of autoencoders using 3 layers, 4 layers, and 5 layers. Figure 15 shows the feature vector extraction effect using 3-layer, 4-layer, and 5-layer autoencoders. Figure 16 shows the ARI coefficients of the clustering results corresponding to the 10 initial threshold parameters. 

As shown in Figure 14, the training time for the three different layer types of autoencoder gradually increases. In rare cases, the time to train a 4-layer autoencoder will be similar to the time to train a 3-layer autoencoder. The time to train a 5-layer autoencoder is generally longer than that of a 4-layer autoencoder. After calculation, the average training time of 3-layer, 4-layer, and 5-layer autoencoders is 142.6 s, 175.05 s, and 218.85 s, respectively. The training time of the 4-layer autoencoder is 22.76% longer than that of the 3-layer autoencoder. The training time of the 5-layer autoencoder is 25.02% longer than that of the 4-layer autoencoder. From the perspective of training time and better experimental results, using a 4-layer autoencoder is a compromise choice.

As can be seen from Figure 15, among the three types of autoencoders, the cosine similarity between the input payload vector and the decoder output vector of the 4-layer autoencoder and the 5-layer autoencoder is generally higher than that of the 3-layer autoencoder. It shows that the feature extraction ability of the original input vector of the 4-layer and 5-layer autoencoders is higher than that of the 3-layer autoencoder. After calculation, the cosine similarities of the payload vectors of the 3-layer, 4-layer, and 5-layer autoencoders are 0.44, 0.49, and 0.51, respectively. The cosine similarity of the 4-layer autoencoder is improved by 11.49% compared to the 3-layer, and that of the 5-layer is improved by 5.37% compared to the 4-layer. Through comparison, it is found that the effect of feature vector extraction in the 4-layer autoencoder on the dataset of this article is better than that of the 3-layer autoencoder. Although the cosine similarity of the 5-layer autoencoder is higher than that of the 4-layer autoencoder, the improvement is not as great as that over the 3-layer autoencoder. Considering the training time of the autoencoder, it is more appropriate to use a 4-layer autoencoder.

From Figure 16, it can be observed that using an initial threshold parameter of 1.1545 for hierarchical clustering yields a higher ARI. The proposed method dynamically adjusts the threshold parameter for hierarchical clustering over five consecutive iterations to achieve more accurate clustering results. The strategy for each dynamic adjustment is outlined in Section 4.2 where the threshold parameter is increased by 0.0005 at each step. Therefore, theoretically, setting the initial threshold parameter to 1.1520 could yield results similar to those obtained with a parameter of 1.1545. Figure 16 also shows that when the initial threshold parameter falls within the range of [1.1520, 1.1545], the ARI generally fluctuates between approximately 0.28 and 0.325. Selecting 1.1545 as the initial threshold parameter enables faster convergence to an optimal clustering result during the five iterations of dynamic parameter adjustment.

### 4.4. Experimental Evaluation of Different Classifier Models 

Evaluation metrics: The performance of various classifier models in generating the payload state predictor on the dataset in this paper was evaluated using Precision [27], Recall [27], and F1-Score [28].


**A3: Comparative Evaluation Experiment (Answering RQ3)**


The essence of the payload state predictor is a prediction task implemented using a classifier model. In order to demonstrate the criticality and importance of the payload state predictor for the prediction accuracy of the prediction scheme, this section designs a comparative experiment and uses multiple evaluation metrics to evaluate multiple prediction effects of various classifier models, CNN, SVM, Naive Bayes, K−neighbour, and Random Forest. The training data are communication payload vectors. We used different classifier models to train on the training set, and used the classifier generated by training to predict the state type of the payload vector on the test set data. The results were compared with the correct state type labels to obtain experimental data.

Experimental results and analysis: Table 2 shows the comparison of prediction effects using different classifier models on the experimental dataset of this article.

As can be seen from Table 1, the effect of payload state prediction using SVM is the best on the Mozi botnet dataset, with an accuracy of 91% and an average of 12% better than other classifier models; the recall rate reached 78%, and the F1-Score value reached 0.84. Among them, CNN is a deep learning model and requires a large amount of data for training; the botnet communication payload dataset of this article is small, so CNN does not work well on this dataset. Through experimental data, it is found that using SVM as the state predictor of payloads can achieve a more accurate prediction effect of payload state types.

### 4.5. Experimental Evaluation of Prediction Schemes

Evaluation metrics: We used the metrics in Section 4.4 to evaluate the effectiveness of the prediction scheme for network state, and evaluated the execution efficiency of different schemes in terms of training time and prediction time.


**A4: Comparative Evaluation Experiment (Answering RQ4)**


This section compares the proposed scheme with other approaches for predicting network payload state. By replicating reverse engineering methods for network protocols, such as NetZob [5,9], we used a method that reads the relevant state fields from the payload and determines its position within a constructed state machine to predict the state. The three different schemes were trained and tested on the same dataset, and their training time, prediction time, and performance metrics were evaluated.

Experimental results and analysis: Table 3 shows the comparison of prediction effects using different schemes on the experimental dataset of this article.

As can be seen from Table 3, the prediction effect of the prediction scheme for payload state in an IoT network combined with the cluster analysis and state machine similarity in this article has good effectiveness metrics on the Mozi botnet dataset, but the execution efficiency is average. NetZob [11] continuously constructs the state of the protocol state machine by actively communicating with the target node during the running process and uses the inferred protocol format to reverse the network communication protocol. If prior knowledge of protocol fields is missing, its inference effect is not good, and the state cannot be effectively extracted from the payload. Moreover, because it uses active communication to construct the state machine, it needs to communicate with real network nodes. The time for network interaction is included, so the corresponding training time and prediction time are longer. The scheme [9] analyzes the communication payload based on the multi-sequence comparison method, and then predicts the state by judging the state field in the payload. The training time and prediction time are shorter than for NetZob [11] and IoTguard, but due to the state prediction completely relying on the inference of the communication protocol format, the effectiveness of this solution is low. Based on the experimental results, the proposed prediction scheme IoTguard for the state of network payload, based on the similarity between network traffic and protocol state machines, can improve the effectiveness of predicting network payload states while maintaining efficient execution.

## 5. Discussion

The experiments in this paper are based on the Mozi botnet. The reason for choosing the Mozi botnet is that it is representative of IoT networks. Mozi is a typical peer-to-peer (P2P) network, where the nodes serve both as clients and servers, and the network protocol is relatively complex. There are two types of IoT networks: one is P2P-based, and the other has a client–server (CS) structure. The CS structure is simpler than the P2P structure. Therefore, our proposed scheme is applicable not only to P2P networks but also to CS-structured networks.

## 6. Conclusions

This paper designs and proposes a prediction scheme for network payload state which is suitable for unknown IoT network scenarios in the absence of prior knowledge. We construct a predictor, IoTguard, to predict payload states in this article. IoTguard can classify network traffic according to payload state type, train recognizers for each state type of payload, and timeously obtain the state type of network payloads in real-time network traffic. We propose an identification method of a label-free network payload state based on a hierarchical clustering algorithm and similarity of protocol state machine. Experiments show that this solution could improve the prediction accuracy of IoT network payload states and have high accuracy in warning of harmful payloads.

## Figures and Tables

**Figure 1 sensors-25-00073-f001:**
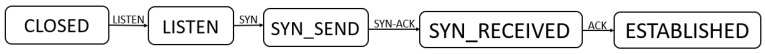
TCP message state machine.

**Figure 2 sensors-25-00073-f002:**
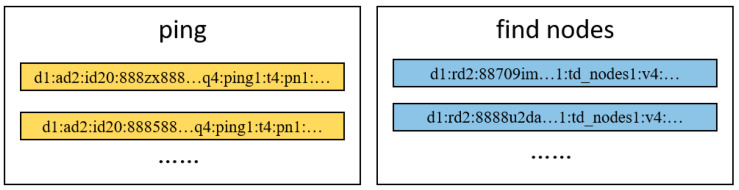
The two clusters formed after payload state clustering.

**Figure 3 sensors-25-00073-f003:**
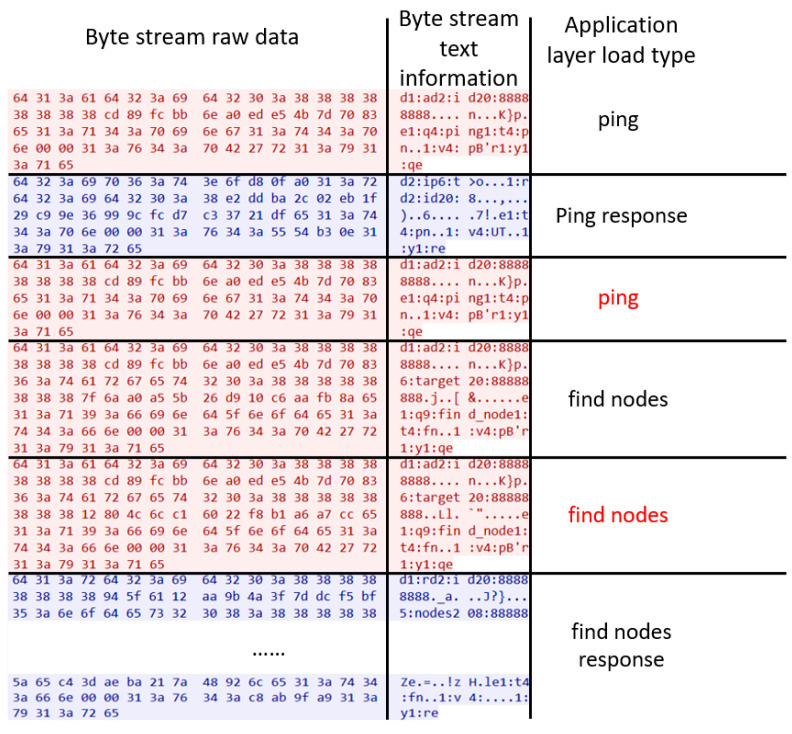
Timing sequence of UDP datagrams from a pair of Mozi zombie nodes. The load type marked in red are out-of-order fields affected by network jitter.

**Figure 4 sensors-25-00073-f004:**
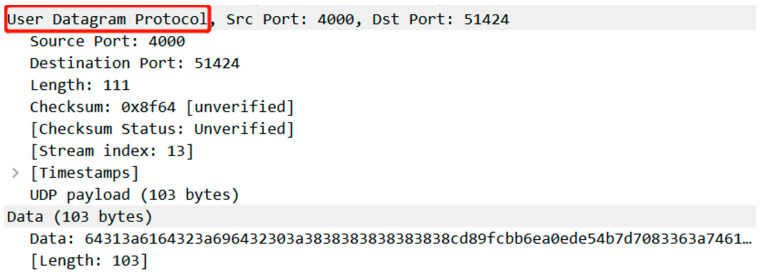
Wireshark parses the Mozi communication payload. The red box is the datagram type UDP packet.

**Figure 5 sensors-25-00073-f005:**
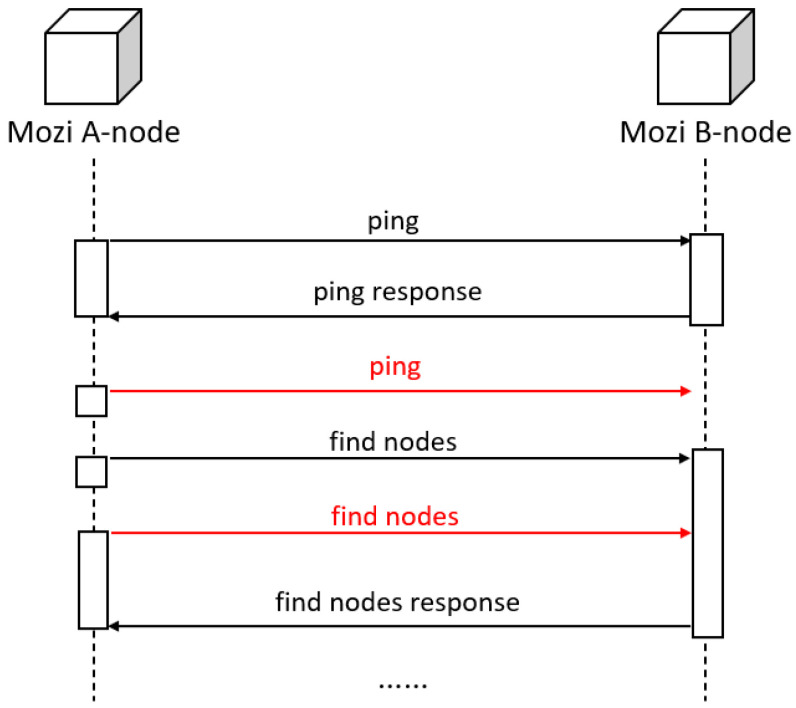
Communication process between Mozi nodes when the network is poor.

**Figure 6 sensors-25-00073-f006:**
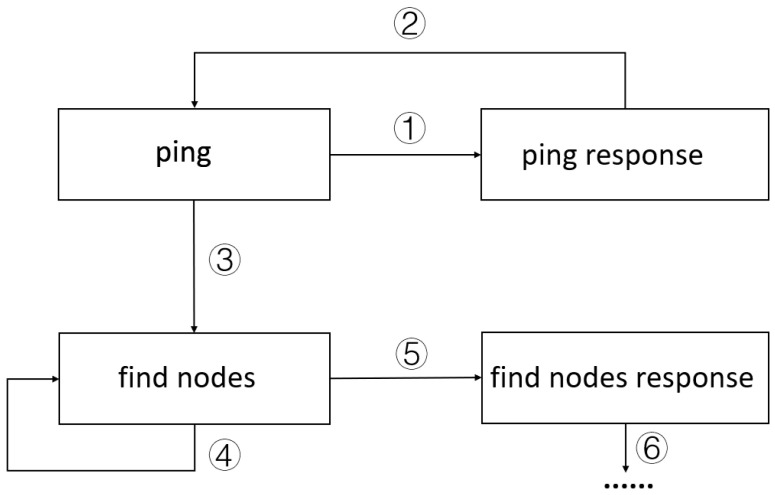
State machine generated by bad network conditions. The serial numbers in the diagram label the state transitions in chronological order.

**Figure 7 sensors-25-00073-f007:**
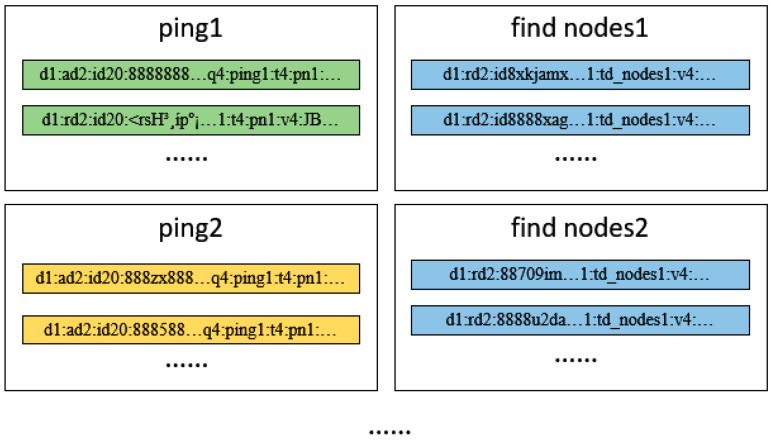
Mozi communication payload clustering results.

**Figure 8 sensors-25-00073-f008:**
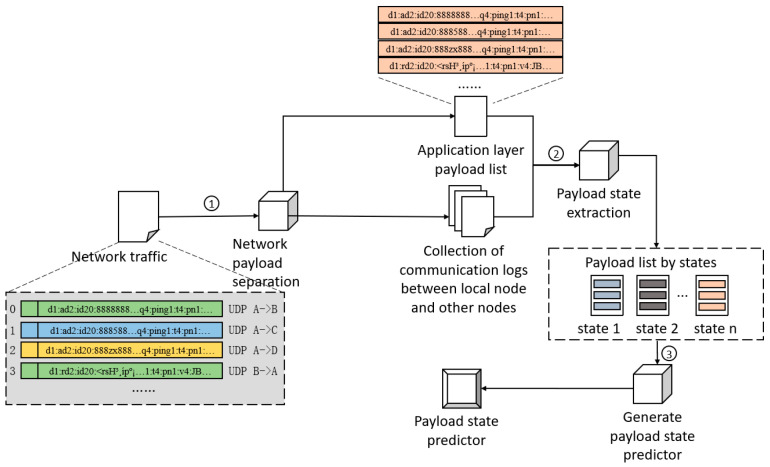
P2P network payload status prediction scheme framework. The serial numbers in the figure mark the steps in chronological order.

**Figure 9 sensors-25-00073-f009:**
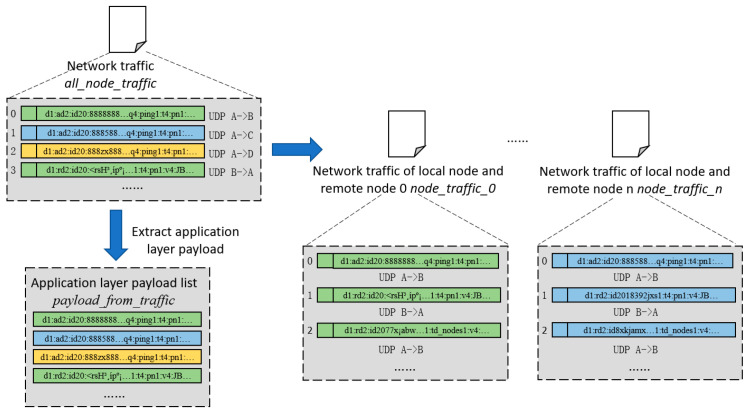
Application layer network payload extraction process.

**Figure 10 sensors-25-00073-f010:**
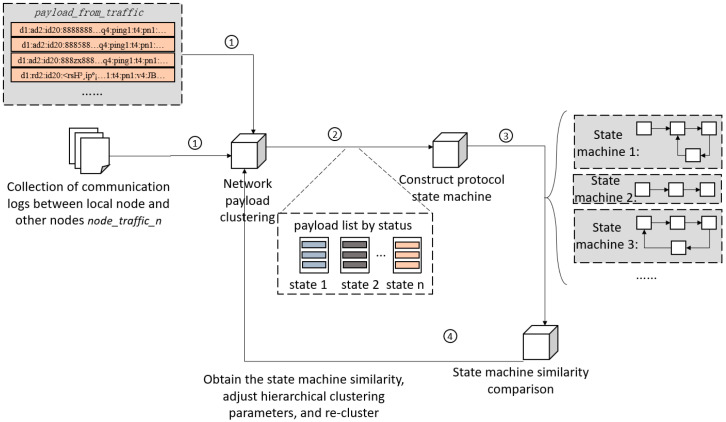
Payload status extraction process. The serial numbers in the figure mark the steps in chronological order.

**Figure 11 sensors-25-00073-f011:**
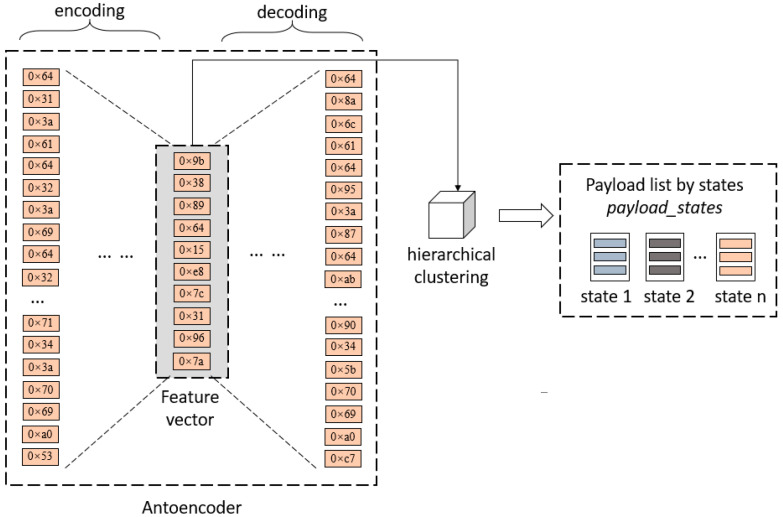
Payload clustering process.

**Figure 12 sensors-25-00073-f012:**
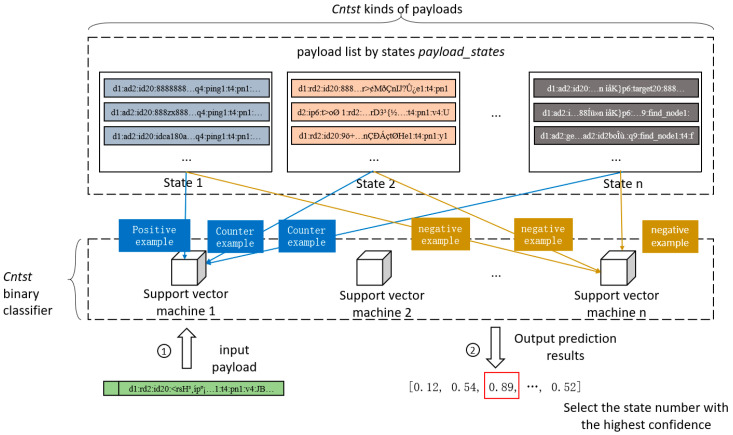
Process of generating payload status predictor. The serial numbers in the figure mark the steps in chronological order.

**Figure 13 sensors-25-00073-f013:**
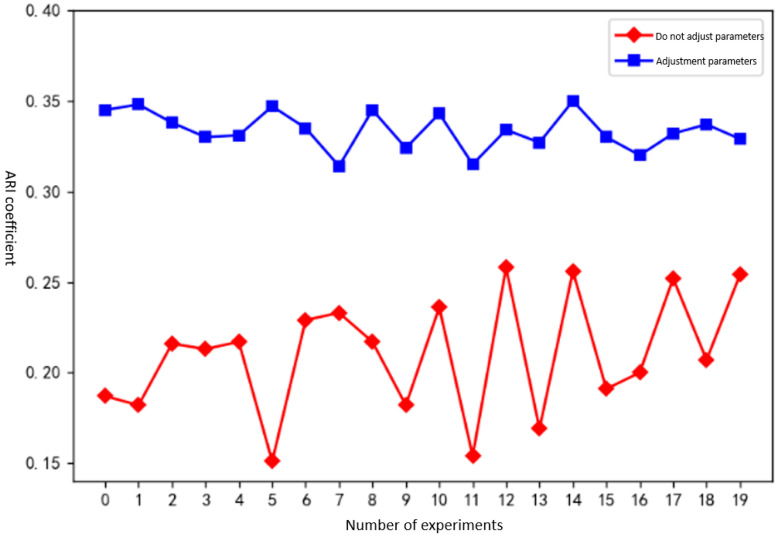
Clustering effect of adjusting the relationship between hierarchical clustering parameters and ARI coefficients.

**Figure 14 sensors-25-00073-f014:**
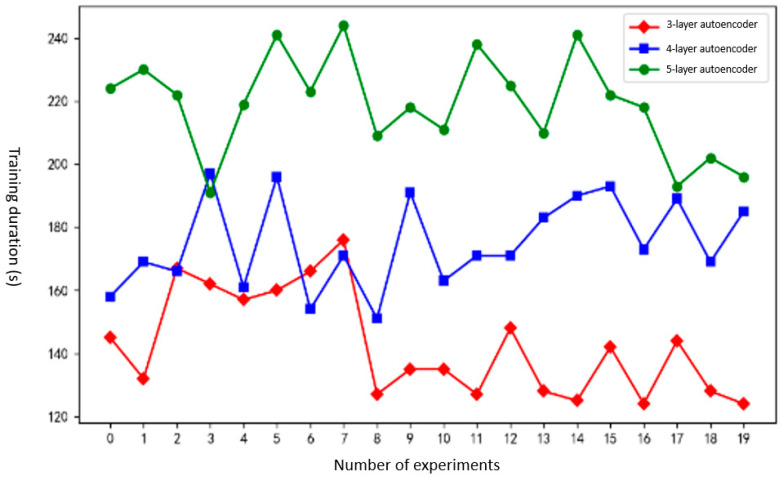
Corresponding training time using autoencoders with different layers.

**Figure 15 sensors-25-00073-f015:**
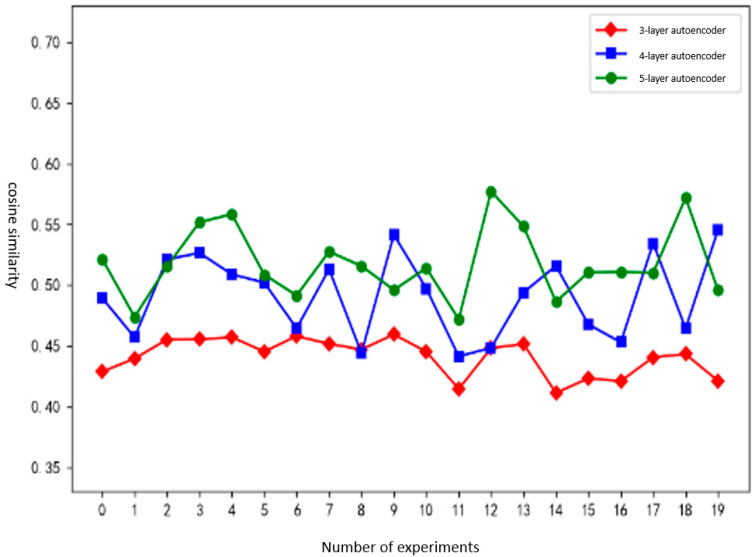
The effect of feature vector extraction using autoencoders with different layers.

**Figure 16 sensors-25-00073-f016:**
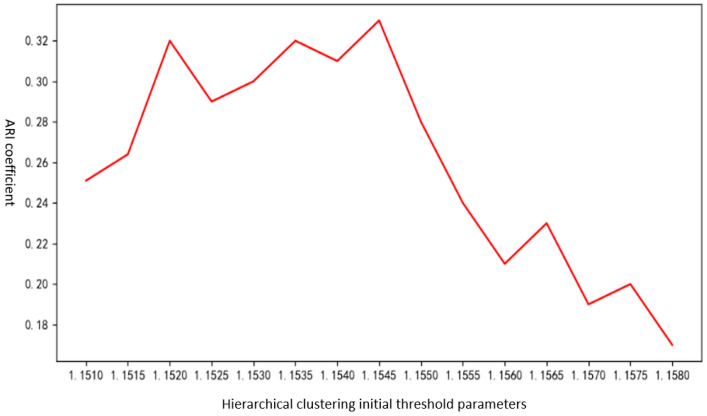
Relationship between initial parameters of hierarchical clustering and ARI coefficient.

**Table 1 sensors-25-00073-t001:** Autoencoder hyperparameters.

Level	Enter Dimensions	Activation Function
1	32	do not use
2	64	Relu
3	128	Relu
4	256	Relu
5	512	Relu

**Table 2 sensors-25-00073-t002:** Comparative experimental results of different classifier models.

Classifier Model	Precision	Recall (%)	F1-Score
CNN	56	32	0.41
SVM	91	78	0.84
Naive Bayes	87	65	0.74
KNN	84	71	0.77
Random Forest	88	72	0.79

**Table 3 sensors-25-00073-t003:** Comparative experimental results of different prediction schemes for network protocol state.

Method	Precision (%)	Recall (%)	F1-Score	Training Time (s)	Prediction Time (s)
NetZob [11]	78	82	0.80	536.19	41.73
Meng et al. [9]	65	73	0.69	109.73	<1
IoTguard	86	81	0.83	252.97	2.26

## Data Availability

The original contributions presented in the study are included in the article, further inquiries can be directed to the corresponding author.

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
