# Peer review of "Payload State Prediction Based on Real-Time IoT Network Traffic Using Hierarchical Clustering with Iterative Optimization"

_sensors, 2024, doi:10.3390/s25010073_

Round 1
Reviewer 1 Report
Comments and Suggestions for Authors
The topic of the paper sounds relevant to the field. It is technically and scientifically sound. Some comments need to be clarified before publication:
Throughout the paper, the authors mention different aspects of the Mozi botnet, and this seems to be the main scope of the paper: identifying malicious traffic in an IoT network compromised by a Mozi botnet. First, I recommend that the authors introduce the motivations and challenges of dealing with Mozi botnets somewhere in the introduction, maybe after the second paragraph. Second, the authors should reference the dataset used for experimentation. And finally, perhaps you should add a discussion about the applicability of your solution to other types of attacks in IoT networks.
Why did you choose to use hierarchical clustering? What are the expected output groups of the clustering algorithm?
The state of the start is short. I would expect authors to contrast their work with others' works in the field, for example, by performing a qualitative comparison.
In Algorithm 2, where do you change the values of state_trans? Maybe I am missing something.
Are the results obtained from a simulation or a real deployment? In both cases, I would expect more details on implementation.
In Table 3, which is your solution?
Improve the quality of figures in the experimentation section.
Please proofread the paper to correct any typos. For example, line 299, 306, 621.
Reviewer 2 Report
Comments and Suggestions for Authors
The paper introduces and explores a novel payload states prediction paradigm. The proposed predictor can predict the payload states in IoT network. The predictor is trained in payload set and some experimental results using Mozi botnet data are given.
After reading the paper, I have some doubts about what the authors bring as contribution and what is the added value of the proposal compared to related work. Although the authors state their contributions in prediction topic, this aspect is not well indicated evident in section 2.
In the Introduction of the paper, the authors should enumerate all sections of the paper to facilitate quicker understanding. The following comments should be revised:
1. It is not clear why enumerating AI algorithms in subsection 2.1 without giving used references
2. Please add more references on paragraph 2.1.1 for better understanding
3. Please add tiem complexity analysis for the proposed algorithms
4. The problem formulation is not explicitly illustrated. Please add a section or subsection to describe the problem formulation and threat model in this paper.
5. The paper needs more results to show the effectiveness of the proposal by adding FP (Faulse Positive) and FN results for example
6. Proofread is suggested for the paper to correct a few typos
7. Most references are outdated, please add a few recent works
Comments on the Quality of English Language
6. Proofread is suggested for the paper to correct a few typos
Reviewer 3 Report
Comments and Suggestions for Authors
Provide a clearer explanation of the gap addressed by IoTGuard compared to other payload prediction methods beyond Netbot.
Include a justification for the 4-layer autoencoder as the optimal choice in Table 1.
Consider providing examples of payload clustering results for better reader comprehension.
Enhance the labeling and description of Figures 5 and 6 to clarify their relevance to the discussion on state machines.
Add a paragraph on the practical implications of deploying IoTGuard in real-world IoT networks.
Round 2
Reviewer 1 Report
Comments and Suggestions for Authors
Authors have attended all my commentaries. I consider that the paper is in shape for publication.
Just some minor:
Line 42: typecal -> typical